Molecular and morphological congruence of three new cryptic Neopetrosia spp. in the Caribbean

http://orcid.org/0000-0003-4741-0261 Vicente Jan 1 janv@hawaii.edu
Ríos Jaime Andrés 2
Zea Sven 3
http://orcid.org/0000-0001-6339-4340 Toonen Robert J. 1
1 University of Hawai‘i at Mānoa, Hawai‘i Institute of Marine Biology , Kāne‘ohe, HI , USA
2 Universidad Nacional de Colombia—Sede Bogotá—Departamento de Biología , Ciudad Universitaria, Bogotá , Colombia
3 Universidad Nacional de Colombia—Sede Caribe—Instituto de Estudios en Ciencias del Mar–CECIMAR, c/o INVEMAR, Rodadero Sur , Playa Salguero, Santa Marta , Colombia
Pawlik Joseph
Electronic publication date: 2019 Feb 5
Publication date: 2019
Volume: 7
Electronic Location ID: e6371
Received 2018 Oct 25; Accepted 2018 Dec 28
Copyright: © 2019 Vicente et al.
Copyright year: 2019
Copyright holder: Vicente et al.
License: This is an open access article distributed under the terms of the Creative Commons Attribution License, which permits unrestricted use, distribution, reproduction and adaptation in any medium and for any purpose provided that it is properly attributed. For attribution, the original author(s), title, publication source (PeerJ) and either DOI or URL of the article must be cited.
License URL: https://creativecommons.org/licenses/by/4.0/

Keywords: Demosponges, Haplosclerida, Neopetrosia, Molecular systematics, Caribbean

Funding: Colombian Administrative Department of Science and Technology—COLCIENCIAS CO–30003–33–81, 30003–154–83, 2105–09–030–86, 2105–09–023–93 US National Science Foundation—NSF INT–86117–17 INVEMAR 220–50, 220–54, 220–95, and others Universidad Nacional de Colombia—Bogotá Campus CINDEC 006–1982 and others Caribbean Campus HERMES 26594 Universidad Nacional de Colombia Caribbean Campus Program for Undergraduate Special Academic Practice 40000008163, with Laboratory and bench work carried out at INVEMAR facilities in Santa Marta, under agreement with Universidad Nacional de Colombia Smithsonian Tropical Research Institute short-term fellowship NSF Postdoctoral Research Fellowship in Biology 2023254002 Smithsonian Tropical Research Institute CNRS France NSF-OA#1416889 Taxonomical studies of sponges from the Colombian Caribbean have been carried out as part of several ecological, chemical, and systematic studies funded by the Colombian Administrative Department of Science and Technology—COLCIENCIAS (grants CO–30003–33–81, 30003–154–83, 2105–09–030–86, 2105–09–023–93), the US National Science Foundation—NSF (grant INT–86117–17), INVEMAR (grants 220–50, 220–54, 220–95, and others), and Universidad Nacional de Colombia—Bogotá Campus (grants CINDEC 006–1982 and others) and Caribbean Campus (grant HERMES 26594). The work of Jaime Andrés Ríos at Santa Marta was funded by Universidad Nacional de Colombia Caribbean Campus Program for Undergraduate Special Academic Practice (code 40000008163), with Laboratory and bench work carried out at INVEMAR facilities in Santa Marta, under agreement with Universidad Nacional de Colombia. Project funding for collection of type material in Bocas del Toro was provided to Jan Vicente through a Smithsonian Tropical Research Institute short-term fellowship. Stipend money to work on histology and sequencing of type specimens was provided to Jan Vicente by NSF Postdoctoral Research Fellowship in Biology (award no. 2023254002). Analyses of sponges from Panama by Sven Zea were carried out as part of the “PorToL Integrative Taxonomy Workshop 2012” at Bocas del Toro, funded by the Smithsonian Tropical Research Institute. Those from Martinique were studied by Sven Zea during the “2013 Training Course on the sponge biodiversity of the Caribbean Sea, workshop of La Martinique” and the “Kick-off meeting of the Associated International Laboratory MARRIO,” both funded by CNRS France. Support for sequencing of sponge samples and publication was provided by the NSF-OA#1416889 grant to Robert J. Toonen. The funders had no role in study design, data collection and analysis, decision to publish, or preparation of the manuscript.

==============================
Neopetrosia proxima (Porifera: Demospongiae: Haplosclerida) is described as a morphologically variable sponge common on shallow reefs of the Caribbean. However, the range of morphological and reproductive variation within putative N. proxima led us to hypothesize that such variability may be indicative of cryptic species rather than plasticity. Using DNA sequences and morphological characters we confirmed the presence of three previously undescribed species of Neopetrosia. Morphological differences of each new congener were best resolved by partial gene sequences of the mitochondrial cytochrome oxidase subunit 1 over nuclear ones (18S rRNA and 28S rRNA). Several new characters for Neopetrosia were revealed by each new species. For example, N. dendrocrevacea sp. nov. and N. cristata sp. nov. showed the presence of grooves on the surface of the sponge body that converge at the oscula, and a more disorganized skeleton than previously defined for the genus. N. sigmafera sp. nov. adds the (1) presence of sigma microscleres, (2) significantly wider/longer oxeas (>200 μm), and (3) the presence of parenchymella larvae. Sampling of conspecifics throughout several locations in the Caribbean revealed larger spicules in habitats closer to the continental shelf than those in remote island locations. Our study highlights the importance of integrating molecular and morphological systematics for the discrimination of new Neopetrosia spp. despite belonging to one of several polyphyletic groups (families, genera) within the current definition of the order Haplosclerida.

Introduction

Cryptic species have posed a challenge to taxonomy and biodiversity studies for over 300 years, but access to DNA sequencing has provided relatively simple tools to resolve species boundaries among morphologically similar species (Bickford et al., 2007; Stat et al., 2012). Particularly for taxa belonging to highly diverse orders with variable growth forms and limited morphological characters, such as corals and sponges, the integration of molecular and morphological approaches can be invaluable (Wörheide & Erpenbeck, 2007; Concepcion et al., 2008; Forsman et al., 2009). In sponges, congruence of molecular and morphological datasets have been successful at the subclass level and have reclassified Demospongiae into subclasses Verongimorpha, Keratosa, and the Heteroscleromorpha (Borchiellini et al., 2004; Sperling, Peterson & Pisani, 2009; Cárdenas, Pérez & Boury-Esnault, 2012; Morrow & Cárdenas, 2015). The presence of siliceous megascleres (monaxons and/or tetraxons) and highly diversified microscleres as synapomorphic characters in Heteroscleromorpha were substantiated by partial nuclear gene sequences (28S rRNA and 18S rRNA) and mitochondrial gene sequence (Holmes & Blanch, 2007; Lavrov, Wang & Kelly, 2008). However, congruence of morphological and molecular datasets for lower taxonomic classifications within Heteroscleromorpha (>6,800 species) have been unsuccessful. Most species within Heteroscleromorpha belong to the order Haplosclerida (1,101 species) (Morrow & Cárdenas, 2015; Van Soest et al., 2018). Although mitochondrial and nuclear genes show Haplosclerida to form a well-supported divergent clade from Heteroscleromorpha (Lavrov, Wang & Kelly, 2008; Thacker et al., 2013), almost every family within Haplosclerida is polyphyletic (Redmond et al., 2011, 2013).

Among these polyphyletic families is the Petrosiidae, which currently consists of 212 species with most of these belonging to Petrosia Vosmaer, 1885 (120 species), followed by Xestospongia De Laubenfels, 1932 (57 species), Neopetrosia De Laubenfels 1949 (33 species), and Acanthostrongylophora Hooper, 1984 (two species) (Van Soest et al., 2018). Xestospongia and Neopetrosia are mainly distinguished on the basis of spicule size, the former usually having spicules larger than 200 μm and the latter shorter. Neopetrosia congeners are distributed worldwide and nine are found in the Tropical Western Atlantic. These include Neopetrosia carbonaria Lamarck, 1814, N. subtriangularis Duchassaing De Fonbressin, 1850, N. proxima Duchassaing De Fonbressin & Michelotti, 1864, N. rosariensis Zea & Rützler, 1983, N. dominicana Pulitzer-Finali, 1986, and N. sulcata Santos, Sandes, Cabral & Pinheiro, 2016, which are found in shallow to deep reefs; and N. dutchi Van Soest, Meesters & Becking, 2014, N. eurystomata Van Soest, Meesters & Becking, 2014, and N. ovata Van Soest, Meesters & Becking, 2014, which are recently discovered mesophotic reef species. Mitochondrial and nuclear sequence data have been published for eight congeners which deeply diverge from one another and are polyphyletic (Redmond et al., 2011; Thacker et al., 2013; Redmond et al., 2013). Mindful of the polyphyletic nature of Neopetrosia, our purpose for this study was not to find markers that resolve the monophyly for this genus but rather use a pairwise comparison of mitochondrial and nuclear DNA sequences of our material with those from GenBank to confirm molecular and morphological separation for new congeners in the Caribbean.

Among tropical W. Atlantic Neopetrosia, N. proxima is a rather widespread species, distributed from the Bahamas to Northern Brazil and shows considerable habitat and geographical variability (Zea, 1987; Zea, Henkel & Pawlik, 2014). In fact, detailed morphological revision of material previously considered to belong to this species has yielded new species (Santos et al., 2016). In this study, Santos and colleagues distinguished N. sulcata from N. proxima by noticing a digitate morphology, lighter color tones with no differentiation between the ectosome and choanosome. While reviewing material of what was believed to be N. proxima or close relatives from Colombia, Panamá, and Martinique, we found several morphologically distinct morphotypes. After detailed molecular barcoding with partial sequence of the cytochrome oxidase subunit 1 (COI), 28S rRNA, and 18S rRNA and morphological comparisons, we were able to distinguish three new species from morphologically similar N. proxima, which we describe and compare here.

Materials and Methods

Specimen collection

Sponges were photographed in situ and collected in Bocas del Toro-Panamá, Colombia and Martinique at depths ranging between 4 and 36 m. Specimens from Colombia were collected at Golfo de Urabá, Cartagena and Santa Marta on the South American coast, and the San Andrés/Old Providence Archipelago in the SW Caribbean. Field observations (in vivo) of each specimen’s morphology, color, consistency, surface, oscules, exudates, and odors were recorded. Samples were preserved in 95% ethanol, and 4% paraformaldehyde (PFA) for histological examination. Samples preserved in PFA for 2–3 days were later transferred to 70% ethanol.

Type and other specimens were deposited in the Florida Museum of Natural History (catalogue number beginning with acronym UF) in Florida, USA, the Makuriwa Museum of Marine Natural History of Colombia at the Institute of Marine and Coastal Research—INVEMAR in Santa Marta (acronym INV POR) and the Natural Science Institute at the National University of Colombia in Bogota (acronym ICN-MHN(Po)). Fragments were also deposited in the Zoological Museum of Amsterdam at the Naturalis Biodiversity Center in Leiden, The Netherlands (acronym ZMA.POR). Fragments of specimens collected in Panamá were deposited in the Museum of Marine Biology and Limnology at the University of Panamá as required by the collection permit of fauna Nr. 5 issued by the “Autoridad Nacional del Ambiente.” Collecting in Colombia was carried out under Decree 309–2003 of the Ministry of the Environment and Sustainable Development as part of the ongoing project “Sponges of the Colombian Caribbean” of INVEMAR’s Makuriwa Museum. Some uncatalogued samples were studied during the “Porifera Tree of Life Project Workshop” in Bocas del Toro, Panamá, August 2012. Uncatalogued samples from Martinique were studied during the “2013 Training Course on the sponge biodiversity of the Caribbean Sea, workshop of La Martinique” and the “Kick-off meeting of the Associated International Laboratory MARRIO” in December 2013 (see also Pérez et al., 2017).

DNA extraction, sequencing, and phylogenetic analysis

Sponge pieces (30 mg) were removed from type material (preserved in 95% ethanol) collected in Panamá (UF 3854, UF 3856–3860) and were used for DNA extractions. DNA was extracted using the Promega E.Z.N.A. Tissue DNA Kit, following the manufacturer’s instructions. DNA concentrations were checked by absorbance ratios using a UV–visible spectrophotometer (Thermo Scientific NanoDrop, Wilmington, DE, USA). DNA from the first elution was diluted to a working stock concentration of 35 ng μL−1.

A list of primers for polymerase chain reaction (PCR) amplification targeting fragments of the COI, the D1–D2 region of the 28S rRNA gene sequence, and the 18S rRNA gene sequence are provided in Table S1. Partial sequences of the different Neopetrosia spp. were made possible using previously reported primer combinations in our PCR reactions (Folmer et al., 1994; Kelly-Borges & Pomponi, 1994; Geller et al., 2013; Chombard, Boury-Esnault & Tillier, 1998). Specific primers were then designed using NetPrimer (http://www.premierbiosoft.com/netprimer/netprlaunch/netprlaunch.html) when sequence data was missing to complete the gene sequence region of interest.

Polymerase chain reactions were carried out in 25 μL total volume including the following: nine μL of H2O, 12.5 μL of BioMix™ Red (Bioline, Taunton, MA, USA) PCR Mastermix, 0.5 μL of each primer (10 mM), two μL of BSA (100 μg/mL), and 0.5 μL of template DNA. The PCR program consisted of an initial denaturation at 94 °C for 3 min followed by 35 cycles of 94 °C for 30 s, annealing temperatures ranged between 45 and 60 °C for 30 s to 1 min 30 s depending on the primer combination and gene product of interest, and 1 min extension at 72 °C. A final extension at 72 °C for 8 min finished the reaction. Primer combinations and annealing temperature for each PCR product is listed in Table S2. PCR products were all ran on a 1% agarose gel stained with GelRed and purified using EXOFAP (EXO1 and FastAP). Sequencing reactions were performed using the BigDye TM terminator v. 3.1, and sequencing was done with an ABI Prism 3730XL automated sequencer.

Forward and reverse reads were sequenced to achieve the greatest base calling accuracy for each species and targeted gene fragment. Sequence chromatograms in forward and reverse directions were trimmed (at an error probability limit of 0.05). Chromatograms were then assembled and edited by eye using Geneious 10 (Kearse et al., 2012). Base calling while editing was made using the highest confidence score for any given base on one of the two chromatograms. All assembled chromatograms resulted in >90% high quality base pair reads with a mean Phred quality score ≥40. Assembled sequences were saved and exported as a fasta file. Each fasta file from targeted gene sequences and each Neopetrosia spp. were checked for contamination using the BLAST (Altschul et al., 1990) function from GenBank. BLAST results that showed >85% sequence identity and a query cover of >60% to those belonging to Porifera were exported to Geneious 10 and aligned using the ClustalW function with default parameters. Alignments were generated using 439 bp of the COI gene sequence, 821 bp of the D1–D2 region of the 28S rRNA and 638 bp for the 18S rRNA gene sequence. Phylogenetic trees were rooted on outgroups Baikalospongia intermedia DAQ167168.1, Axinella corrugata KC869523.1 and EF092264.1 for COI, 28S rRNA and 18S rRNA, respectively. A plugin for MrBayes version 3.2.1 (Huelsenbeck & Ronquist, 2001) for Bayesian inference (BI) and RaxML (Stamatakis, 2006) was added for phylogenetic analyses using a maximum likelihood (ML) framework in Geneious 10. Both analyses were implemented using the GTRGAMMA model with 1,000 bootstrap replicates. The BI was run using 5 million generations sampled every 200 generations. The analysis was stopped when the standard deviation (SD) of split frequencies fell below 0.01. At this point convergence was assumed and the burnin value was determined. Phylogenetic trees were generated in Mega7 (Kumar, Stecher & Tamura, 2016). Resulting bootstrap values of >50 from the ML and Bayesian posterior probabilities >0.50 from the BI analysis were incorporated to the tree. Sequences of holotypes and other specimens for each species collected in Panamá were deposited in GenBank and assigned accession numbers reported in Table S3.

Sectioning and spicule preparation

Permanent slides with clean spicules and thick (∼1 mm) histological sections (tangential and perpendicular) were prepared for each specimen following the methods in Zea (1987). Spicules were digested from small (20 mg) sponge pieces soaked in commercial sodium hypochlorite and shaken for 12 h. Spicules were subsequently washed and centrifuged three times with DI and resuspended in ethanol; a few drops of spicule suspensions were added to microscope slides, dried on a warm plate, and mounted on Permount®. Tissue sections were either dried on a warm plate or dehydrated and stained in successively stronger ethanol solutions (96%, 100%), and then cleared in xylene; then sections were mounted on Permount®. Individual spicule types and skeletal framework were photographed with a Zeiss AxioCam ERc5s mounted on a Zeiss AxioLabA.1 light microscope. Photographs were processed in Photoshop and measurements carried out from photos with AxioVision SE64 Rel.4.9.1 and ImageJ® (Abràmoff, Magalhães & Ram, 2005) (http://imagej.nih.gov/ij/). The lengths and widths of 50 spicules per specimen and spicule types are presented as (minimum–mean (±1 SD)–maximum length/width in μm). A few drops of the spicule suspension from Panamanian specimens were added to a stub, air dried, and imaged under high vacuum with a JEOL 5600 SEM scanning electron microscope (SEM) at the Nano Imaging Facility, University of Maryland Baltimore County. Spicule suspension from Colombian and Martinique specimens were carbon coated with a Quorum Q150R and photographed under a QUANTA 200 FEI SEM. Measurements of spicule tracts, skeletal arrangement of fibers, and meshes were compared across species and specimens from different collection sites.

The electronic version of this article in portable document format will represent a published work according to the International Commission on Zoological Nomenclature (ICZN), and hence the new names contained in the electronic version are effectively published under that Code from the electronic edition alone. This published work and the nomenclatural acts it contains have been registered in ZooBank, the online registration system for the ICZN. The ZooBank LSIDs (Life Science Identifiers) can be resolved and the associated information viewed through any standard web browser by appending the LSID to the prefix http://zoobank.org/. The LSID for this publication is: (urn:lsid:zoobank.org:pub:B56217F8-FA57-4D93-8A69-BEC98F4B2AE7). The online version of this work is archived and available from the following digital repositories: PeerJ, PubMed Central, and CLOCKSS.

Results

Phylogenetic analysis

The phylogenetic relationship between novel Neopetrosia spp. using mitochondrial (COI) and nuclear genes (28S rRNA/18S rRNA) reconfirmed the polyphyletic nature of this genus (Erpenbeck et al., 2007; Redmond et al., 2011; Setiawan, 2014; Setiawan et al., 2018) (Fig. 1). Nevertheless, the use of different markers allowed us to detect enough genetic differences across all N. proxima paratypes and new Neopetrosia spp. In particular, COI showed the highest resolution of sequence dissimilarity between all new congeners and confirmed our hypothesis that morphological variability was indicative of cryptic species (Fig. 1A). For example, N. proxima, “N. dendrocrevacea sp. nov.,” and “N. cristata sp. nov.” were all closely related and formed a divergent clade that was closely related (87% identical) to Amphimedon queenslandica Hooper & Van Soest, 2006 sequence EU237474.1 (Kayal & Lavrov, 2008). Within this clade all N. proxima morphotypes were 100% identical to each other, 96% identical to “N. dendrocrevacea sp. nov.,” 95% identical to “N. cristata sp. nov.,” and 81% identical to “N. sigmafera sp. nov.”. “N. sigmafera sp. nov.” was the most distantly related (<85% sequence similarity) congener with a well-supported and deeply divergent clade. The closest relative to “N. sigmafera sp. nov.” was Gelliodes wilsoni Carballo, Aquilar-Camacho, Knapp & Bell, 2013 with 99% identity. Additional congeners like N. exigua Kirkpatrick, 1900 sequence KX454496.1 and N. seriata Hentschel, 1912 sequence JN242213.1, were distantly related (<85%) from all new congeners in this study.

Figure 1 Phylogenetic trees.

Bayesian and maximum likelihood topology generated from partial sequences spanning the (A) Folmer (5′) region of the cox1 gene, (B) D1–D2 region of the 28S rRNA gene, and (C) 18S rRNA gene, from Haplosclerida taxa generated in this study (blue) and sequences downloaded from GenBank. Clades in (C) correspond to clades assigned by Redmond et al. (2013). Sequences in bold highlight other Neopetrosia species. Bootstrap values less than 50% have been omitted from the trees. Numerical values at nodes show Bayesian posterior probabilities followed by RAxML bootstrap values. Nodes with “—” refer to the absence of the node generated by RAxML. Type specimens are indicated by an asterisk at the end of the specimen’s catalogue number.

The phylogenetic tree of the 28S rRNA gene showed a very similar topology to COI. For example, as in COI, a well-supported clade (1/100) with all specimens of N. proxima, “N. dendrocrevacea sp. nov.,” and “N. cristata sp. nov.,” were also deeply divergent from the rest of the congeners in the tree (Fig. 1B). Nonetheless, 28S rRNA did show a lower resolution of sequence dissimilarity, with no sequence differences between “N. cristata sp. nov.” and “N. dendrocrevacea sp. nov.” Both sequences of these species were 97% identical, with strong support (1/100), to all N. proxima morphotypes in the clade. All congeneric sequences in this clade were 93–94% identical to Petrosia lignosa Wilson, 1925 KC869595.1. In addition, “N. sigmafera sp. nov.” showed <85% sequence identity to all congeners in this clade. The closest relatives to “N. sigmafera sp. nov.” were Gelliodes callista De Laubenfels, 1954 KC869562.1 (89% identical) and Xestospongia deweerdtae Lehnert & Van Soest, 1999 KX668524.1 (90% identical). Additional sequences from congeners like N. rosariensis Zea & Rützler, 1983 KC869457.1 and N. subtriangularis KC869591.1, were <85% identical to all new species. The closest congener to “N. sigmafera sp. nov.” appears to be N. carbonaria with 88% sequence identity.

The phylogeny of Neopetrosia spp. using 18S rRNA resulted in the lowest resolution of sequence dissimilarity with four congeners being 100% identical and grouping into Clade C (Redmond et al., 2011) (Fig. 1C). Identical congeners include “N. cristata sp. nov.,” N. proxima, N. exigua, and “N. dendrocrevacea sp. nov.”. The sequence from “N. sigmafera sp. nov.” was 93% identical to congeners in Clade C, and grouped into Clade B, which also included N. carbonaria. N. rosariensis grouped into Clade A and was 95% identical to congeners in Clade C, and 97% identical to “N. sigmafera sp. nov.”.

Systematics

Class Demospongiae Sollas, 1885

Subclass Heteroscleromorpha Cárdenas, Pérez & Boury-Esnault, 2012

Order Haplosclerida Topsent, 1928

Family Petrosiidae Van Soest, 1980

Definition. Haplosclerida with an ectosomal skeleton consisting of an isotropic reticulation of single spicules or spicule tracts and choanosomal skeleton verging towards an isotropic reticulation of spicule tracts, in which primary and secondary tracts are indistinct (Van Soest, 1980).

Genus NeopetrosiaDe Laubenfels, 1949

Definition. Petrosiidae with finely hispid surface produced by fine brushes of oxeas issued from subectosomal tracts, and a compact choanosomal network combining rounded meshes with a superimposed anisotropic reticulation. Megascleres oxeas <200 μm long (Desqueyroux-Faúndez & Valentine, 2002).

Neopetrosia proximaDuchassaing De Fonbressin & Michelotti, 1864

(Fig. 2; Fig. S1; Table 1)

Figure 2 Neopetrosia proxima Duchassaing De Fonbressin & Michelotti, 1864.

In situ images of Panama specimens (A) UF 3856, (B) UF 3858, and ex situ image of (C) UF 3860, with corresponding images of (D–F) tangential sections of the ectosome (LM); (G–I) perpendicular sections through the ectosome and choanosome (LM); (J–L) size and morphological variations of oxeas (SEM).

Table 1 Spicule measurements of oxeas (length and width) of Neopetrosia spp. described in this study.

Species	Specimen	Location	Length (μm)	Width (μm)	
Neopetrosia proxima
Duchassaing De Fonbressin & Michelotti (1864)	UF 3856	Bocas del Toro, Panama	98–158.7 (±19.9)–193	3–10.9 (±1.7)–11	
UF 3858	Bocas del Toro, Panama	92–146.6 (±14.0)–168	3–9.0 (±1.7)–12	
UF 3860	Bocas del Toro, Panama	117–159.3 (±14.0)–181	6–9.3 (±1.1)–12	
Uncat. PPA 35	Bocas del Toro, Panama	75–NA (± NA)–205	2.7–NA (± NA)–10.7	
Uncat. PPA 37	Bocas del Toro, Panama	85–NA (± NA)–167	1.7–NA (± NA)–12.7	
INV POR1306	Old Providence, Colombia	110–NA (± NA)–150	2.5–NA (± NA)–5	
Neopetrosia dendrocrevacea sp. nov.	UF 3854 Holotype	Bocas del Toro, Panama	91–165.2 (±15.9)–188	2.8–7.4 (±1.5)–10.5	
Uncat. PPA 07	Bocas del Toro, Panama	111—156.5 (±14.6)—181	4.5—6.6 (±0.9)—8.9	
INV POR0535	Urabá, Colombia	134–171.4 (±12.9)–198	4.4–7.4 (±1.0)–9.4	
INV POR 1336	Cartagena, Colombia	133–165.1 (±12.9)–189	4.6–6.4 (±0.6)–7.7	
INV POR1337	Cartagena, Colombia	139–164.4 (±12.9)–192	4.1–7.3 (±1.3)–9.8	
ICN-MHN(Po) 0269	Santa Marta, Colombia	130–151.6 (±9.4)–168	5.0–6.5 (±0.9)–9	
INV POR1335	Santa Marta, Colombia	103–147.8 (±12.3)–169	4–6.5 (±1.1)–9	
INV POR1333	San Andrés, Colombia	86–119.3 (±11.3)–150	3.5–4.8 (±0.6)–6	
INV POR1334	San Andrés, Colombia	114–130.1 (±7.7)–149	5.1–3.8 (±0.6)–7	
Neopetrosia cristata sp. nov.	UF 3859 Holotype	Bocas del Toro, Panama	121–142.1 (±9.8)–163.2	2.1–7.2 (±1.7)–9.6	
Neopetrosia sigmafera sp. nov.	UF 3857 Holotype	Bocas del Toro, Panama	173–235.9 (±14.1)–259	6.5–13.6 (±1.5)–15.9	
Uncat. PPA 36	Bocas del Toro, Panama	174–226.5 (±12.4)–248	6.5–14.2 (±2.4)–17.6	
Uncat. PPA 48	Bocas del Toro, Panama	196–232.7 (±10.4)–233	6.9–13.5 (±1.9)–15.5	
INV POR 1338	Cartagena, Colombia	203.5–237.4 (±11.8)–255	5.8–14.0 (±2.8)–14.0	
INV POR 1339	Cartagena, Colombia	201.9–240.2 (±14.5)–260	5.0–17.0 (±2.9)–17.7	
ICN-MHN(Po) 270	Cartagena, Colombia	201.9–240.2 (±14.5)–260	5.0–17.0 (±2.9)–17.7	
Uncat. SZ-20	Martinique	153–197.2 (±10.6)–219	6.9–8.2 (±0.6)–9.3	
Uncat. SZ-21	Martinique	115–204.6 (±18.6)–230	6.8–8.8 (±0.7)–10.3	
Uncat. SZ-23	Martinique	130–190.4 (±13.2)–190	4.9–6.8 (±1.0)–8.6	
Notes:

Measurements are expressed as minimum–mean (±1 standard deviation)–maximum. N = 50.

NA, not available; Uncat, uncatalogued sample.

Thalysias proxima Duchassaing De Fonbressin & Michelotti, 1864: 84, Pl. VIII, Figs. 2 and 3.

Densa araminta De Laubenfels, 1934: 14.

Neofibularia proxima; Wiedenmayer, 1977: 147, 255.

Xestospongia proxima; Van Soest et al. 1983: 198; Van Soest, 1984: 143; Zea, 1987: 116, Fig. 34, pl. IX, Figs. 3 and 4; Van Soest & Stentoft 1988: 132, pl. XII, Fig. 4, text, fig. 64; Lehnert & Van Soest 1996: 77, Fig. 29; Díaz, 2005: 470; Collin et al. 2005: 648; Rützler et al. 2000: 278.

Neopetrosia proxima; Campos et al. 2005: 13, Figs. 8A–D; Muricy et al. 2011: 106 (with further synonyms from Brazil); Zea, Henkel & Pawlik, 2014 (field guide); Santos et al. 2016: 336, Fig. 4; Van Soest, 2017: 35, Fig. 21a–d; Pérez et al. 2017: 10.

Material examined. Bocas del Toro, Panamá: UF 3856, Punta Caracol (9.3777° N, 82.1265° W) eight m in depth, coll. Jan Vicente, May 8, 2015; UF 3858 and UF 3860 Dolphin Rock (9.35076° N, 82.1863° W), 14 m in depth coll. Jan Vicente and Arcadio Castillo May 20, 2015. Uncatalogued fragments PPA 35, 37 and 38, Isla Colón, Aeropuerto (9.3339° N, 82.2548° W), on rubble and sand, fringing reef, seven m in depth, coll. Sven Zea, August 9, 2012. Colombia: INV POR1304, Bahía de Santa Marta, El Morro (11.2494° N, 74.2302° W) reef base, 30–36 m in depth, coll. S. Zea, February 10, 1988. INV POR1306, San Andrés Archipelago Old Providence, north of Low Cay (Pallat Bank, 13.5525° N, 81.3245° W), fore reef terrace, 25 m in depth, coll. Sven Zea, October 19, 1994. Further Colombian material is described in Zea (1987).

Description (Fig. 1; Table 1). The external morphology varies from cylindrical (Fig. 2A) or flat branching individuals (from 5 × 15 cm by 5 cm thick), to thickly encrusting (two cm thick) mounds (Figs. 2B and 2C); encrusting specimens often fill cavities and appear level with the substratum. Oscule size varies between two and seven mm in diameter and are either randomly scattered along the body of the sponge (Figs. 2A and 2C), or aligned along elevated ridges (Fig. 2B). A white membrane collar surrounding the oscules was observed in some individuals (Fig. 2C). Consistency is toughly compressible but difficult to cut with a scalpel or a knife. The surface texture is velvety, from even and smooth (Fig. 2A) to rugged (Figs. 2B and 2C), often knobby from conical or blunt elevations around oscules; massive specimens often have keyhole to irregular grooves. All individuals produced a sticky substance when cut or squeezed in situ. Surface color across individuals from Panamá varied from yellow (Fig. 2A), dark brown (Fig. 2B), to light purple (Fig. 2C); Santa Marta specimens in Colombia are characteristically violet to pink (see Zea, 1987); in other areas color is predominantly yellowish to purplish dark brown. Internal coloration across all specimens is light-yellow.

Skeleton. The skeleton consists of a fasciculated reticulation of isotropic multispicular tracts that form circular to irregularly elongated meshes. In the ectosome, a paratangential reticulation of tracts (20–200 μm) makes meshes that vary between 120 and 400 μm in diameter (Figs. 2D–2F; Figs. S1A–S1C) depending on the individual (180–300 μm (Fig. 2D), 80–240 μm (Fig. 2E), and 280–390 μm (Fig. 2F)). Smaller circular meshes in the ectosome seem to be the result of thicker spicule tracts (80–170 μm, Fig. 2E; Fig. S1B), when compared to individuals with thinner spicule tracts (50–100 μm, Figs. 2D–2F; Figs. S1A–S1C). Dark purple pigments from cyanobacteria penetrate about 750 μm into the choanosome (Figs. 2G and 2H; Figs. S1D and S1E). In some individuals, pigments were not observed from the surface but 500 μm below the ectosome (Fig. 2I; Fig. S1F). The ectosome can also be distinguished by the presence of large (500 μm) subectosomal spaces, clearly visible in some individuals (Figs. 2G and 2I; Figs. S1D and S1F), but in others it forms smaller (250 μm) openings (Fig. 2H; Fig. S1E) as a result of denser and thicker spicule tracts. Erect ascending spicule brushes radiate at the ectosome surface. The choanosome also shows a large number of circular meshes that vary in abundance and size (200–700 μm) according to the thickness of spicule tracts (Figs. 2G–2I; Figs. S1G–S1I).

Spicules. Most spicules are slightly curved, symmetric oxeas with very few strongyloxeas present (Figs. 2J–2L); some are more curved and there is variation in size with developmental stage. Oxea endings vary between hastate and conical shapes. Size 92–205 μm long by 1.7–12 μm wide (Table 1).

Habitat and ecology. This species is found living from shallow rocky shores and reefs, to deep reef habitats in a variety of wave-exposures (Zea, 1987; Zea, Henkel & Pawlik, 2014); also, in caves (Pérez et al., 2017). Specimens UF 3858 and UF 3860 were collected in a highly exposed reef (Dolphin Rock) with strong wave energy, while specimen UF 3856 was collected inside Almirante Bay (Punta Caracol) with very low wave exposure. Strong wave energy is known to influence the appearance of aligned oscula (observed in X. deweerdtae collected in the same site, see Fig. 7B of Vicente, Zea & Hill, 2016) and is apparent in specimen UF 3856. Brooding larvae were not observed in any specimens; zoanthids were also absent.

Distribution. Bahamas (Zea, Henkel & Pawlik, 2014). Caribbean: Puerto Rico, US Virgin Islands, Jamaica, Martinique, Barbados, Panamá, Colombia, Belize (Zea, 1987; Van Soest & Stentoft, 1988; Lehnert & Van Soest, 1996; Rützler et al., 2000; Díaz, 2005; Collin et al., 2005; Zea, Henkel & Pawlik, 2014; Pérez et al., 2017). Guyana (Van Soest, 2017). Brazil: North to North East Regions (Amapa, Maranhão, Rio Grande do Norte and Sergipe states) (Campos et al., 2005; Muricy et al., 2011; Santos et al., 2016).

Taxonomic remarks. All N. proxima specimens collected in this study exhibited varied morphologies (physical appearance, color, thickness of fiber tracts, circular meshes). These differences initially lead us to think that these were heterospecific. However, these variations showed no nuclear or mitochondrial genetic differences, and seem to be plastic characters within this species. Upon closer examination, spicule sizes, spicule shapes, the skeletal arrangement of the choanosome and ectosome are all in agreement with previous descriptions (Zea, 1987; Díaz, 2005; Zea, Henkel & Pawlik, 2014).

Neopetrosiadendrocrevacea sp. nov.

(Fig. 3; Fig. S2; Table 1)

Figure 3 Neopetrosia dendrocrevacea sp. nov.

In situ images of individual PPA 07 (A) and the holotype UF 3854 (B), both from Panama, and of the paratype ICN-MHN(Po) 0269 (C) from Santa Marta, Colombia, with corresponding (second and third rows) images of (D–F) tangential sections of the ectosome (LM); (G–I) perpendicular sections through the ectosome and choanosome (LM); size and morphological variations of oxeas from specimens collected in (J) Uraba, (K) Panama, (L) Cartagena, (M) Santa Marta, and (N) San Andrés Archipelago (LM). Scale bar of (D) is 100 μm.

Haplosclerida unident. sp. 1; Zea 2001, Table 1.

Neopetrosia sp. –“soft”; Zea, Henkel & Pawlik, 2014 (field guide).

?Neopetrosia proxima; Zea, Henkel & Pawlik, 2014 (field guide, in part, only two images of partly branching and knobby individuals, taken in Panamá, Bocas del Toro, Isla Solarte, Punta Hospital, March 3, 2012, identified from fresh spicule preparations).

Type material and type locality. Holotype: UF 3854, Panamá, Bocas del Toro, STRI point (9.3429° N, 82.1258° W), two m in depth, coll. Jan Vicente, June 10, 2015. Paratypes: Colombia: ICN-MHN(Po) 0269, Bahía de Nenguange, playa del Manglar, Santa Marta (11.2494° N, 74.2301° W), 1.5 m in depth, coll. Sven Zea, March 18, 1999. INV POR1335, Bahía de Chengue (11.3200° N, 74.1267° W), 1.5 m in depth, coll. Sven Zea, May 19, 1982. INV POR1336, Bancos de Salmedina, Cartagena (10.3735° N, 75.6663° W), 24 m in depth, coll. Sven Zea, August 19, 1980. INV POR1337, Islas del Rosario, Isla Rosario (10.1583° N, 75.8050° W), eight m in depth, coll. Sven Zea, March 7, 1998. INV POR0535, Cabo Tiburón, Golfo de Urabá (8.6840° N, 77.3710° W), nine m in depth, coll. Sven Zea, September 28, 1995. INV POR1333, Isla de Providencia, San Andrés Archipelago (13.5058° N, 81.3558° W), 16 m in depth, coll. Sven Zea, October 21, 1994. INV POR 1334, Banco Serrana, leeward terrace, San Andrés Archipelago (14.4592° N, 80.2740° W), 16 m in depth, coll. Sven Zea, May 14, 1995.

Additional material. Bocas del Toro, Panamá: uncatalogued sample PPA 07, Isla Bastimentos, Adriana’s reef (9.2419° N, 82.1736° W), five m in depth, coll. Sven Zea, March 2, 2012.

Description. Thin to thick (one cm) encrustations growing up to 30 cm in diameter; or made up of coalescing, one to two mm thick branches, elevating to 10–15 cm from the base (Fig. 3B). The surface has densely reticulated or scattered characteristic grooves that converge at the rim of the oscules, cutting through them and making them appear lumpy or incomplete (Figs. 3A–3C); sometimes the grooves surround smooth knobs of varied sizes. Oscular diameter range from one to two mm in encrusting individuals to 0.5 cm in branching ones. A translucent membrane surrounds the oscules, sometimes closing them. Consistency from slightly soft to firm, but crumbly. Texture is particularly velvety and when squeezed in situ the sponge produces a sticky substance. External color is golden yellow to reddish brown to dark purple with ochre yellow tinges; light-yellow in ethanol. Interior color light-yellow.

Skeleton. Ectosome as a paratangential reticulation, composed of rather confused, loose, uni to paucispicular tracts, up to 4–10 spicules and 25–70 μm across, forming polygonal meshes 100–200 μm in diameter (Figs. 3D–3F; Fig. S2A–S2C). Single spicules and spicule brushes from the end of choanosomal ascending tracts pierce the surface. Pigments from cyanobacteria penetrate about 600 μm inside the choanosome. The choanosome consists of an anisotropic reticulation with distinguishable, but loose primary tracts, 6–13 spicules and 10–50 μm across, separated by 50–200 μm (Figs. 3G–3I; Figs. S2D–S2F). Tracts are interconnected by solitary spicules or loose paucispicular tracts, forming confused meshes measuring 80–300 μm in diameter (Figs. 3G–3I; Figs. S2G–S2I).

Spicules. Symmetric oxeas, curved, with hastate endings (short but thick pointed ends, 86–198 μm long by 2.8–10.5 μm wide (Table 1). Spicule sizes vary by geographic location. For example, spicules from specimens collected closer to the continental shelf (i.e., Urabá) measured 171.4 ± 12.9 μm × 7.4 ± 1.0 μm while those collected on the insular shelf (i.e., San Andrés Archipelago) were smaller and measured 130.1 ± 7.7 μm × 3.8 ± 0.6 μm (Table 1; Figs. 3J–3N).

Habitat and ecology. This species is found on shallow rocky substrates (1.5 m) and deep reefs (16 m), living on dead coral rubble or over other sponges. This species is a common sponge of the leeward fore reef terrace of Banco Serrana in the San Andrés Archipelago with an average density of 0.56 individuals per 20 m2 (Zea, 2001).

Distribution. Panamá (Bocas del Toro), Colombia (Urabá, Cartagena, Santa Marta, San Andrés Archipelago, cf. Zea, 2001), Puerto Rico (Zea, Henkel & Pawlik, 2014). S.Z. examined a dried fragment from the Bay of Honduras which belongs to this species (courtesy of J.C. Lang).

Taxonomic remarks. Although some specimens initially analyzed showed different characteristics from N. proxima, like Haplosclerida unident. sp. 1 (Zea, 2001), or as Neopetrosia sp.-“soft” (Zea, Henkel & Pawlik, 2014), others were thought to be N. proxima (e.g., ICN-MHN(Po) 0269 and INV POR1335). Accordingly, a more detailed molecular and morphological analysis was pursued to detect less obvious differences. COI sequence data of N. dendrocrevacea sp. nov. was 96% identical to N. proxima and confirmed heterospecificity to N. proxima (Fig. 1). Some obvious morphological differences between these species lie in the consistency of individuals, where N. proxima is generally firmer and tougher to cut than N. dendrocrevacea sp. nov. N. proxima also exudes a stickier mucus when cut. Oscules are larger in N. proxima and the surface lacks the grooves that seem to be a diagnostic morphological character of N. dendrocrevacea sp. nov. The arrangement of the choanosomal and ectosomal skeleton shows very distinct morphologies from N. proxima, with reticulation being more isotropic in N. proxima. Meshes are also larger in diameter and better organized in N. proxima; multispicular tracts are thicker, more dense and fasciculated as described by Campos et al. (2005) and Zea (1987). In the field, N. dendrocrevacea sp. nov. can be easily confused with Svenzea cristinae Alvarez, Van Soest & Rützler, 2002 which is also a crumbly, thin to thicker encrustation with yellow tinges, but its spicules are long styles (Zea, Henkel & Pawlik, 2014). N. dendrocrevacea sp. nov. also shares some similar external features with Haliclona (Soestella) walentinae Díaz, Thacker, Rützler, Piantoni, 2007 including the sometimes bumpy surface between shallow grooves, and the similar oxea (100–180 × 3–9 μm). The latter are more thinly encrusting and soft, has a looser and more unispicular skeleton, and the tissue is crisscrossed by purple filamentous cyanobacteria.

Etymology. The given species name is an adjective derived from the Greek word dendron that refers to tree, and crevace from the old French word referring to groove (Brown, 1956) which denotes the presence of branching and meandering grooves along the surface of the sponge. We use the feminine dendrocrevacea assuming that Neopetrosia is feminine, following Article 31.2 of the International Code for Zoological Nomenclature (http://www.iczn.org/, accessed on October 1, 2018).

Neopetrosia cristata sp. nov.

(Fig. 4; Table 1)

Figure 4 Neopetrosia cristata sp. nov.

Holotype (UF3859) (A) ex situ image of live sponge specimen; (B) tangential section of the ectosome (LM); (C) perpendicular section through the ectosome and choanosome (LM); (D) close-up of perpendicular section through the ectosome (LM); (E) close-up of perpendicular section through the choanosome (LM); (F) variation of oxeas (SEM).

Type material and type locality. Holotype: UF 3859, Panamá, Bocas del Toro, Dolphin Rock (9.35076° N, 82.1863° W), 14 m in depth, coll. Jan Vicente and Arcadio Castillo, May 20, 2015.

Description. The holotype is a thickly (up to one cm) encrusting sponge with an irregular shape, 10 cm in diameter. Surface with scattered pointy conulose ends or smooth ridges. Oscules aligned on ridges along the sponge body, sometimes on top of conical elevations, <1 mm in diameter. There are also sometimes narrow grooves that converge around oscules (Fig. 4A, arrow). Consistency is firm but crumbly when torn. Surface texture is smooth and velvety. Specimens exude a sticky substance when squeezed in situ. External color is reddish brown to dark purple and the interior is light-yellow. Interior and exterior tissues turned to a light-yellow color in ethanol.

Skeleton. The ectosome is composed of a rather confused reticulation of loose multispicular tracts, 3–15 spicules and 40–120 μm across, forming circular to polygonal meshes, 150–250 μm in diameter (Fig. 4B). Cyanobacterial pigments penetrate to 700 μm inside the choanosome (Fig. 4C). The choanosome consists of a confused reticulation of loose multispicular tracts, 5–20 spicules and 60–100 μm across (Fig. 4D), forming circular meshes, 100–150 μm in diameter (Fig. 3E).

Spicules. Slightly curved oxeas, 121–160 × 2.1–9.6 μm (Fig. 3F; Table 1).

Habitat and ecology. The holotype was found in a spur and groove, high wave energy environment, growing on a dead coral skeleton.

Distribution. Bocas del Toro, Panamá.

Taxonomic remarks. As predicted by similarities in their morphological characters, the COI phylogeny showed N. cristata sp. nov. to be more closely related to N. dendrocrevacea sp. nov. than any other congeneric, with 98% identity, and diverged from N. proxima with strong bootstrap support (Fig. 1A). Additional sequence data spanning the COI I3-M11 extension (700 bp product alignment) showed that N. dendrocrevacea sp. nov. and N. cristata sp. nov. were 96% identical which further supports their heterospecificity. This species shares many external morphological characters with N. dendrocrevacea sp. nov. These characters are (1) the appearance of grooves along the sponge’s surface that converge at the oscules, (2) the velvety texture of the sponge surface, and (3) the disorganized reticulation of the choanosome and ectosome. Nevertheless, both of these species are distinguishable based on the morphology of the grooves along the surface of the sponge which are a lot less pronounced and fewer in number in N. cristata sp. nov. (Fig. 4A). In N. dendrocrevacea sp. nov. up to seven grooves converge around the oscules in both branching (Fig. 3B) and encrusting (Figs. 3A and 3C) individuals, forming a star-like pattern around the oscules. The appearance of a crown or irregular mounds around the oscules is also missing in N. cristata sp. nov. The diameter of the oscules is <1 mm in N. cristata sp. nov., being larger than one mm in N. dendrocrevacea sp. nov. The surface of N. cristata sp. nov. is also smoother and lacks the rounded knobs surrounded by grooves found in N. dendrocrevacea sp. nov., while those are pointed and dispersed in N. cristata sp. nov. Spicules in Panamá are also somewhat smaller and straighter in N. cristata sp. nov. (holotype UF 3859: 121–142.1 (±9.8)–163.2) than in N. dendrocrevacea sp. nov. (PPA 07: 111–156.5 (±14.6)–181; holotype UF 3854: 91–165.2 (± 15.9)–188).

Etymology. The given species name is an adjective derived from the Latin word crista, referring to the surface ridges of the holotype (Brown, 1956). We use the feminine cristata, assuming that Neopetrosia is feminine, following Article 31.2 of the International Code for Zoological Nomenclature (http://www.iczn.org/, accessed on October 1, 2018).

Neopetrosia sigmafera sp. nov.

(Fig. 5; Fig. S3; Tables 1 and 2)

Figure 5 Neopetrosia sigmafera sp. nov.

In situ images of the holotype UF 3857 (A) and individuals PPA 38 (B), both from Panama, and SZ-21 from Martinique (C), with corresponding images (D) of zoanthids; (E and F) tangential sections of the ectosome (LM); (G) brooding larvae (arrows) (H and I) perpendicular sections through the ectosome and choanosome (LM); Size and morphological variations of oxeas and sigmas from (J) Panama, (K) Cartagena, (L) Martinique (LM).

Table 2 Lengths of sigma of Neopetrosia sigmafera sp. nov.

Specimen	Location	Number of sigmas	Length (μm)	
UF 3857, holotype	Bocas del Toro, Panama	20	7.6–22.0–27.0	
Uncat PPA 36	Bocas del Toro, Panama	15	12.7–22.6–29.3	
Uncat PPA 38	Bocas del Toro, Panama	13	20.6–24.4–29.5	
INV POR 1338	Cartagena, Colombia	10	11.0–20.2–28.7	
INV POR 1339	Cartagena, Colombia	10	8.7–15.1–30.2	
ICN-MHN(Po) 270	Cartagena, Colombia	10	11.2–19.2–31.4	
Uncat SZ-20	Martinique	10	20.6–24.4–29.5	
Uncat SZ-21	Martinique	6	10.0–21.2–28.7	
Uncat SZ-23	Martinique	10	9.3–19.8–31.0	
Note:

Measurements are expressed as minimum–mean–maximum.

Type material and type locality. Holotype: UF 3857, Bocas del Toro, Panamá, Punta Caracol (9.3777° N, 82.1863° W), three m in depth, coll. Jan Vicente, May 8, 2015. Paratypes: Cartagena, Colombia: ICN-MHN(Po) 270, Islas del Rosario, Pajarales, close to Yohmara islet (10.1779° N, 75.7750° W), five m in depth, coll. Sven Zea, March 10, 2002. INV POR1338, 1339, Isla del del Rosario, Pajarales (lagoon) (10.1780° N, 75.7750° W), four to five m in depth coll. Sven Zea, August 13, 2014.

Additional material. Bocas del Toro, Panamá: uncatalogued samples PPA 36, Isla Colón, Aeropuerto (9.3339° N, 82.2548° W), seven m in depth, coll. Sven Zea, August 9, 2012; PPA 48, Isla Cristobal, Buoy 19 (9.3018° N, 82.2943° W), eight m in depth, coll. Sven Zea, August 15, 2012. Martinique: uncatalogued samples SZ-20, SZ-21, Les Anses d´Arlet, Le Grande Anse, Salomon´s Garden, Northeast point (14.5053° N, 61.0947° W), 11–18 m, coll. Sven Zea, December 5, 2013. Uncatalogued fragment SZ-23, Les Anses d´Arlet, Le Grande Anse, Pointe Legarde, Southeast point (14.4969° N; 65.0897° W), 24 m, coll. Sven Zea, December, 2013.

Description. Group of tubes or chimneys or ramified, erect, anastomosed mounds, reaching 10–30 cm in width and 10–30 cm in height (Figs. 5A–5C). Sponge surface is smooth, but sometimes with horizontal crests (sinuous channels) (Fig. 5C). Surface is also quite porous (0.5–1 mm diameter pores) and in some specimens can be reticulated. Oscules are generally apical and measure two to five mm in some individuals (Fig. 5A) and up to a one to two cm in others (Fig. 5C). A translucent membrane surrounding the oscules was obvious in some individuals. Consistency is firm, rigid and tough to cut with scalpel but brittle once squeezed with considerable force. Unlike the other congeneric species described in this study, the sponge did not exude a sticky substance when squeezed in situ. The exterior color varies between brownish amber, yellow with sporadic brownish-green blotches (Fig. 5A), to crimson with light and dark tones (Fig. 5C). Color at the base and in the interior of the sponge is light-yellow.

Skeleton. The ectosome is partially tangential, isodictyal, with unispicular or paucispicular tracts (one to six spicules and 13–75 μm across) (Figs. 5E and 5F; Figs. S3A–S3C). Spongin was detected in some nodal points in the ectosome where ascending choanosomal tracts connect with perpendicular spicules along the ectosome (Fig. 5E; Fig. S3A). Pigments from cyanobacteria penetrate about one mm into the choanosome (Figs. 5H and 5I; Fig. S3D). The choanosome is an anisotropic reticulation with ascending multispicular tracts (four to eight spicules and 1–80 μm across) and occasional 130–250 μm openings, interconnected by single or loosely arranged spicules (Figs. 5H and 5I; Figs. S3G–S3I). The choanosomal tracts have a larger number of free spicules, are thicker, more confused, and become harder to depict deeper into the choanosome. Tracts become thinner as they ascend towards the ectosome, eventually becoming almost unispicular. Channels in the choanosome have a diameter 0.3–2 mm.

Spicules. Slightly curved oxeas, hastate, with conical to sharp ends, 130–260 × 5–18 μm (Table 1). The mean of oxea sizes vary according to the location where specimens were collected (Figs. 5J–5L). C-shaped sigmas are present in the ectosome and choanosome, 8–21–33 μm in length, showing no variation in size among geographic locations but varied abundance across specimens (Table 2).

Habitat and ecology. This species is found on shallow patch reefs and sand flats (three m) in Bocas del Toro and Islas del Rosario (Cartagena), and deeper reef habitats in Martinique (11–24 m). This species is viviparous, being the only congener observed to brood larvae in the summer months in Bocas del Toro, Panamá (Fig. 5G). A detailed description of the morphology and phototactic swimming behavior of N. sigmafera sp. nov. larvae are described by Collin et al. (2010). At the time, N. sigmafera sp. nov. was misidentified as N. proxima, but the morphological assessment in this study clearly shows a different spicule composition, and skeletal arrangement of the choanosome and ectosome from N. proxima. The presence of larvae seems to be a diagnostic character that also helps distinguish it from N. proxima when both are found living in the same habitat. Zoanthids are occasionally found growing on N. sigmafera sp. nov. but not on N. proxima in shallow reef habitats of Bocas del Toro (Fig. 5D). This sponge is known to harbor a host specific community of the cyanobacteria species Synechococcus spongiarum which produce high amounts of chlorophyll-a (Erwin & Thacker, 2007, 2008).

Distribution. Panamá (Bocas del Toro), Colombia (Cartagena), Martinique (Les Anses d’ Arlet). S. Zea observed specimens in Belize (Carrie Bow Cay and Pelican Cays).

Taxonomic remarks. Similar in situ characters shared between N. sigmafera sp. nov. and N. proxima have made their classification difficult over the last decade; particularly in Bocas del Toro, Panamá, where they are sympatric (Figs. 5A and 2A, respectively). Nevertheless, the phylogenetic analysis of both nuclear and mitochondrial sequence data placed N. sigmafera sp. nov. in a well-supported and deeply divergent clade (<85% sequence similarity) from all other new congeners and N. proxima (Fig. 1). These results were further supported by several morphological differences. For example, a closer look at the spicules of each species revealed that oxeas are longer, thicker, and have more hastate endings in N. sigmafera sp. nov. (130–260 × 5–18 μm) compared to N. proxima (85–223 × 2.4–10 μm). N. sigmafera sp. nov. also has sigmas as microscleres, which are never present in N. proxima or any other Neopetrosia spp. The skeleton is also less dense, with less massive spicule tracts in N. sigmafera sp. nov. Oxeas are also smaller and thinner than observed in N. dendrocrevacea sp. nov. and N. cristata sp. nov. Grooves, which are a diagnostic character of N. dendrocrevacea sp. nov. are also absent in N. sigmafera sp. nov. From all congeners, larvae were only found in N. sigmafera sp. nov., suggesting that viviparity seems to be a diagnostic character of this species.

The external morphology of this species is also similar to N. dominicana, but the ladder has strongyles instead of oxeas, and also lacks sigmas. There are also some similarities with X. caminata (Pulitzer-Finali, 1986), although the oscules are much larger (5–10 mm) and spicules are larger (200–260 × 5–14 μm) in the latter species. In addition to oxeas, the spicule composition of X. caminata also includes strongyles, while sigmas are absent. Additionally, although N. sigmafera sp. nov. also shares a similar branching morphology with N. subtriangularis, the skeleton of N. subtriangularis is much more neatly reticulated with numerous circular channels, denser multispicular tracts, and smaller oxeas (131–181 × 1.6–11.7 μm). X. bocatorensis Díaz, Thacker, Rützler & Piantoni, 2007 also has hastate oxeas and sigmas, but oxeas reach greater lengths (230–320 × 8–15 μm), and sigmas have a smaller length range (10–26 vs. 8–33 μm in N. sigmafera sp. nov.). In addition, X. bocatorensis is a thinly encrusting sponge with a purple signature color from associated Oscillatoria filamentous cyanobacteria dispersed throughout the ectosome and choanosome (Díaz et al., 2007). In contrast, color patterns in N. sigmafera sp. nov. are similar to other congeners, having two distinct colors across the body, brown ectosome from cyanobacteria and light-yellow choanosome.

Etymology. The given species name is an adjective that combines the name of the sigma microsclere with the Greek suffix phero, which translates to “carrying” or “bearing” (Brown, 1956). We use the feminine sigmafera, assuming that Neopetrosia is feminine, following Article 31.2 of the International Code for Zoological Nomenclature (http://www.iczn.org/, accessed on October 1, 2018).

Discussion

Molecular and morphological assessments of putative N. proxima and close relatives sampled from Martinique, and the Southern Caribbean, revealed three new species with a variety of new morphological characters, and a new reproductive strategy for the genus. Differences in morphological characters were mostly resolved by partial sequences of the mitochondrial (COI) gene but less so by nuclear genes (28S rRNA and 18S rRNA). Neopetrosia is defined by having the presence of a hispid surface produced by the rise of subectosomal tracts above a compact choanosomal skeleton composed of circular meshes with anisotropic reticulation of oxeas that are <200 μm in length (Desqueyroux-Faúndez & Valentine, 2002). These characters were well supported by COI, 28S and 18S rRNA sequences in three variable specimens of N. proxima which were 100% identical. Identical sequences across the three N. proxima paratypes support plasticity of morphological variations in color, oscula alignment, the size of circular meshes throughout the choanosome and ectosome, and the thickness of spicule tracts. In a closer examination of the new congeners we have found that N. dendrocrevacea sp. nov. and N. cristata sp. nov. have a more confused ectosomal and choanosomal skeleton, with less obvious circular meshes. These morphological differences were supported by a 4–5% divergence in COI sequences of N. cristata sp. nov. and N. dendrocrevacea sp. nov. to N. proxima. The recently discovered congener from mesophotic reefs in Curacao, N. ovata, also shows a similar confused skeleton organization as N. dendrocrevacea sp. nov. and N. cristata sp. nov. (Van Soest, Meesters & Becking, 2014). In addition, N. sigmafera sp. nov. further deviates from this definition by the presence of sigmas (microscleres), oxeas >200 μm in length, and being the only congener so far known to brood larvae. N. sigmafera sp. nov. was also the most distantly related congener to N. proxima based on mitochondrial and nuclear sequences (<85% sequence identity).

Improved resolution of the COI gene over nuclear genes are in agreement with the phylogeny of other Haplosclerida where mitochondrial genes (including the COI I3–M11 extension) resolved up to 12 well supported subclades of Haliclona spp., while ribosomal sequences only resolve six (Knapp et al., 2015). Similar results were also observed in Tethya spp. where mitochondrial genes resolved up to five supported subclades, while ribosomal sequences supported four (Schaffer, Davy & Bell, 2018). In all phylogenetic trees, N. proxima, N. dendrocrevacea sp. nov., and N. cristata sp. nov. formed a well-supported clade with deep divergence from N. sigmafera sp. nov. These results are congruent with multiple diagnostic morphological characters present in N. sigmafera sp. nov. that are absent in all other congeners (i.e., presence of sigmas and brooding larvae).

Despite these striking differences, and distant genetic relatedness to other congeners, it is difficult to place N. sigmafera sp. nov. in a different genus on the basis of its viviparous nature or presence of sigmas. Other than X. bocatorensis, N. sigmafera sp. nov. is the only other larval brooding Petrosiidae (Collin et al., 2010), which rejects the hypothesis that all Petrosiidae are oviparous (Fromont & Bergquist, 1994; Maldonado & Riesgo, 2009), and shows that viviparity is not a good synapomorphic character (Van Soest & Hooper, 2002). In addition, the only other Petrosiidae with sigmas is also X. bocatorensis (Díaz et al., 2007), which shows that sigmas can be shared across different genera within Petrosiidae. Although being closely related to the genus Gelliodes based on mitochondrial and nuclear markers, N. sigmafera sp. nov. shares no morphological characters with this genus other than the presence of sigmas and oxeas.

Shared morphological characters between N. sigmafera sp. nov. and other Neopetrosia, are the pauci- to multispicular ascending and interconnecting tracts, often ending in spicule brushes which support, when present, a tangential uni- to paucispicular reticulation. The firm consistency, the presence of brown- purple pigments in the ectosome with a light-yellow-colored interior, plus the overall size of the spicules are all characters that support a generic morphological classification for N. sigmafera sp. nov. within Neopetrosia. The presence of sigmoid microscleres is generally not a diagnostic character at the generic level in this family or other Haplosclerid families, and together with viviparity, they are not monophyletic across different taxa by mitochondrial or ribosomal molecular markers (Redmond et al., 2011). Thus, given the current lack of congruence between morphological and molecular classifications of Haplosclerida, we are hesitant to erect a new genus for N. sigmafera sp. nov. Its placement should be considered temporary while more suitable molecular markers showing monophyly for its unique characters are discovered.

Our study also highlights the effect that environmental factors may have on the size of oxeas. Previous studies have shown that spicule morphology can be influenced by hypersilicification as a result of high silica concentrations (Maldonado et al., 1999). Sponges can also produce smaller spicules by living in association with other sponges (Vicente et al., 2014). In this study, higher silica concentrations from terrestrial runoff in habitats closer to the continental shelfs are likely the cause of larger oxeas in both N. dendrocrevacea sp. nov. and N. sigmafera sp. nov. collected in Bocas del Toro, Urabá, Cartagena, and Santa Marta (continental shelf), than in specimens collected in San Andres or Martinique (oceanic islands). Similar variations in spicule sizes have been reported for other species collected in sites with low/high terrestrial runoff (Zea, 1987; DeBiasse & Hellberg, 2015; Vicente, Zea & Hill, 2016; Silva & Zea, 2017).

Despite highlighting the polyphyletic nature of Haplosclerida, applying a multilocus based approach using ribosomal and mitochondrial markers continues to prove as a useful tool in resolving the taxonomy between congeneric species. Recently this approach has been used across a wide taxonomic range of sponges (Erpenbeck et al., 2016; Yang et al., 2017). These methods are useful as a first pass assessment of classification for a wide range diversity of sponges, to be subsequently integrated with morphological systematics. However, in order to understand the evolutionary relationship within Haplosclerida we must continue to focus our research efforts toward finding monophyletic markers by sequencing more genomes from species within different families of Haplosclerida.

Conclusions

We report molecular and morphological congruence of three new Neopetrosia spp. in the Caribbean. Molecular congruence was mostly revealed at the highest resolution by partial sequences of the mitochondrial COI and less by nuclear ones (18S rRNA and 28S rRNA). The most distantly related new congener based on partial COI sequences was N. sigmafera sp. nov., which adds the presence of sigma microscleres, significantly wider/longer oxeas (>200 μm), and the presence of parenchymella larvae to the genus. N. dendrocrevacea sp. nov. and N. cristata sp. nov. were confirmed as sister species based on partial COI sequences and by the shared appearance of a more confused skeletal arrangement, and the presence of grooves on the surface of the sponge body converging to its oscula. Differences in morphological characters from N. proxima were also confirmed by differences in COI sequences. Despite being a polyphyletic genetic marker in Neopetrosia spp., our study shows that the partial COI gene fragment continues to be a useful marker in resolving cryptic species belonging to highly diverse orders with variable growth forms.

Supplemental Information

Supplemental Information 1 COI sequence data for Neopetrosia spp.

Exported fasta file of COI sequences of Neopetrosia spp. Sequence file was generated from forward and reverse sequence chromatograms that were assembled and edited by eye using Geneious 10.

Click here for additional data file.

Supplemental Information 2 28S rRNA sequence data for Neopetrosia spp.

Exported fasta file of 28S rRNA sequences of Neopetrosia spp. Sequence file was generated from forward and reverse sequence chromatograms that were assembled and edited by eye using Geneious 10.

Click here for additional data file.

Supplemental Information 3 18S rRNA Neopetrosia spp. sequence data.

Click here for additional data file.

Supplemental Information 4 Individual spicule measurements for Table 1.

Raw data of each spicule measurement to calculate minimum–mean (±1 standard deviation)–maximum of each voucher sample.

Click here for additional data file.

Supplemental Information 5 Individual sigma spicule measurements for Table 2.

Raw data of each sigma spicule measurement to calculate minimum–mean (±1 standard deviation)–maximum of each voucher sample.

Click here for additional data file.

Supplemental Information 6 List of primers used from previous studies and designed to obtain COI, 28SrRNA and 18S rRNA sequences for Neopetrosia spp. described in this study.

Click here for additional data file.

Supplemental Information 7 Primer pairs and annealing temperatures for each PCR product.

Click here for additional data file.

Supplemental Information 8 NCBI accession numbers for COI, 28S rRNA and 18S rRNA sequences of each species.

Click here for additional data file.

Supplemental Information 9 Close-up images of tangential and perpendicular sections of Neopetrosia proxima.

Close-up images of tangential sections of Panama specimens (A) UF 3856 (B) UF 3858, and (C) UF 3860, with corresponding (from second and third rows) images of (D–F) perpendicular sections through the ectosome and choanosome (LM), and close-up image of the choanosome (G–I) (LM). Scale bar in all images is 300 μm.

Click here for additional data file.

Supplemental Information 10 Close-up images of tangential and perpendicular sections of Neopetrosia dendrocrevacea sp. nov.

Close-up images of tangential sections from individual PPA07 (A) and the holotype UF 3854 (B), and of the paratype ICN-MHN(Po) 0269 (C) from Santa Marta, Colombia, with corresponding (from second and third rows) images of (D–F) perpendicular sections through the ectosome and choanosome (LM), and close-up image of the choanosome (G–I) (LM). Scale bar in all images is 200 μm.

Click here for additional data file.

Supplemental Information 11 Close-up images of tangential and perpendicular sections of Neopetrosia sigmafera sp. nov.

Close-up images of tangential sections from the holotype UF 3857 (A), paratype PPA 38 (B), and paratype SZ-21 from Martinique (C), with corresponding (from second and third rows) images of (D–F) perpendicular sections through the ectosome and choanosome (LM), and close-up image of the choanosome (G–I) (LM). Scale bar in all images is 200 μm.

Click here for additional data file.

We thank Cristina Díaz and Robert W. Thacker for providing helpful discussion points on the taxonomy of Neopetrosia spp. We are indebted to Rachel Collin and Plinio Gondola for hosting J.V. at the Bocas del Toro STRI research station. Arcadio Castillo and Micah J. Marty, are thanked for SCUBA diving assistance. We thank Laszlo Takacs at the NanoImaging Facility, University of Maryland Baltimore, and Inga Conti-Jerpe at the University of North Carolina Wilmington, for help with SEM images. This is HIMB contribution Nr. HIMB contribution Nr. 1750, SOEST contribution Nr. SOEST contribution Nr. 10629, CECIMAR contribution Nr. CECIMAR contribution Nr. 480 and INVEMAR contribution Nr. INVEMAR contribution Nr. 1213.

Additional Information and Declarations

Competing Interests

Author Contributions

Field Study Permissions

Data Availability

New Species Registration

Robert J. Toonen serves as an Academic Editor for PeerJ.

Jan Vicente conceived and designed the experiments, performed the experiments, analyzed the data, contributed reagents/materials/analysis tools, prepared figures and/or tables, authored or reviewed drafts of the paper, approved the final draft.

Jaime Andrés Ríos conceived and designed the experiments, performed the experiments, analyzed the data, contributed reagents/materials/analysis tools, prepared figures and/or tables, authored or reviewed drafts of the paper, approved the final draft.

Sven Zea conceived and designed the experiments, performed the experiments, analyzed the data, contributed reagents/materials/analysis tools, prepared figures and/or tables, authored or reviewed drafts of the paper, approved the final draft.

Robert J. Toonen contributed reagents/materials/analysis tools, authored or reviewed drafts of the paper, approved the final draft, provided additional references and key points to the introduction and discussion of the paper.

The following information was supplied relating to field study approvals (i.e., approving body and any reference numbers):

Fragments of specimens collected in Panama were deposited in the Museum of Marine Biology and Limnology at the University of Panama as required by the collection permit of fauna Nr. 5 issued by the “Autoridad Nacional del Ambiente (ANAM).”

Collecting in Colombia was carried out under Decree 309–2003 of the Ministry of the Environment and Sustainable Development as part of the ongoing project “Sponges of the Colombian Caribbean” of INVEMAR’s Makuriwa Museum.

The following information was supplied regarding data availability:

Raw Sequences and data of spicule measurements are available as Supplementary Files.

All accession numbers are available in Table S3. All sequences were deposited in GenBank under accession numbers: MK105442, MK101127, MK101134, MK105443, MK101128, MK101135, MK105446, MK101131, MK101136, MK105444, MK101129, MK101137, MK105441, MK101126, MK101133, MK105445, MK101130, MK101132.

Fragments of specimens collected in Panama were deposited in the Museum of Marine Biology and Limnology at the University of Panama.

The following information was supplied regarding the registration of a newly described species:

Publication LSID: urn:lsid:zoobank.org:pub:B56217F8-FA57-4D93-8A69-BEC98F4B2AE7;

Neopetrosia De Laubenfels LSID: urn:lsid:zoobank.org:act:2DA83FA8-583B-4576-8CE6-3B7B6DD69A6A;

Neopetrosia dendrocrevacea LSID: urn:lsid:zoobank.org:act:EEDD5E08-036B-4834-808B-DB68A6B0856B;

Neopetrosia sigmafera LSID: urn:lsid:zoobank.org:act:6B6539BD-029E-4FE5-AECD-447AE7ADE95E;

Neopetrosia cristata LSID: urn:lsid:zoobank.org:act:09FF04FC-F830-43A5-B014-138026C6863C.

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
