# Peer review of "Molecular and morphological congruence of three new cryptic Neopetrosia spp. in the Caribbean"

_PeerJ, doi:10.7717/peerj.6371_

## Round 0.1 · original submission · Major Revisions

I now have two reviews from experts in sponge phylogenetics, and both recommend major revisions of this ms. Both reviewers praise the authors for tackling a difficult group of sponges with a combined morphological and molecular approach, but both also wanted more detail on the methods and decisions made by the authors to establish their conclusions.

Reviewer 1 ·

Basic reporting

The MS was well structured and followed normal standards for the discipline. The writing was of a good standard throughout, with only a few typos or minor grammatical errors, which I have detailed in the general comments line-by-line. However, there were instances of overuse of the passive voice (e.g. sequences being “retrieved” and “provided” when referring to amplification and sequencing work done in this study), which made it difficult to understand. I would encourage the use of active voice to enhance clarity here, but at the least, these words should be substituted for something more appropriate.

In line 185, the Genbank Accession numbers are not listed. 18S data for N. sigmafera is not present in the Fasta file. Please add these.

Figures were generally high quality, however Fig 2D is bad resolution and must be changed for publication. In Fig 3A the arrows are too small, and should be made a little larger. In figure 5, the words “28S rRNA” look squashed (very minor but would be good to change if possible).

The literature was well referenced and relevant. However, at line 614, please be specific about the name of the extension, as this is not referred to elsewhere as far as I could see, and cite it (presumably it is Erpenbeck’s I3-M11 partition that the authors mean?). Please also give more details upon the previous work detailed in lines 82-84.

Experimental design

This MS represents good quality, original and meaningful research, tackling taxonomy in the difficult Haplosclerida group. The work carried out was extensive and rigorous, and suitable to address the research at hand. However, there are a few clarifications to the methodology I would like to see:

1) The paragraph starting line 147 describing the primers and PCR conditions is confused and would not be sufficient for replication. The specific use of the primers designed here is not clear – i.e. in what primer pair combinations they were used, given that three new primers are shown for COI and one new primer for 18S. The PCR conditions are not described in sufficient detail here, as a range of annealing temperatures is given for all reactions, rather than detailing the temperature used for each primer pair. This could go in S1 to avoid cluttering the main article.

2) I would like to see details of the quality scores and acceptable quality thresholds utilized for the sequence data; details on how the sequences were edited and assembled are scant.

3) At line 170, how was the value of 85% chosen?

Validity of the findings

I agree with the validity of these new species. However, I am not convinced by the authors’ reasons for rejecting a genus change for N. sigmafera given the molecular, morphological and reproduction data all indicate a divergence from the other members of the genus. The authors explain that neither the presence of sigmas or the viviparity can be used as a monophyletic character, but do not explain why they reject their genetic evidence. What more would be required to push N. sigmafera into a different genus if these things are not enough, and conversely what evidence ties N. sigmafera to the genus?

Additional comments

1) It’s a great shame there is no in-situ photo of Neopetrosia cristata – is there any way for the authors to obtain one?

2) The last sentence of the abstract is confusing, please revise.

3) Line by line comments:

Line 51 – 52: Phrase “were confirmed” repeated
Line 61: “consist” should be “consists”. I also recommend putting a comma after the word “Petrosiidae” to enhance readability
Line 65: Is the citation of the web address necessary?
Line 68 – 73: Is there any reason to repeat the genus name throughout this sentence?
Line 74-76: This sentence is unclear, please clarify.
Line 75: Replace “has” with “have”
Line 81: “Rather widespread” is a little general – please be specific about the geographic range
Line 86: Why have you put “other Caribbean localities”? It seems the only other area sampled was Martinique? Sorry if I missed something.
Line 94: “in” should be “on”
Line 130-131: should be “50 spicules per specimen”?
Line 150: The use of the word “retrieved” here is not appropriate (I assume that you mean you amplified these genes using primers from the cited papers?)
Line 151: “Specific primers were then designed when sequence data was missing to complete the gene sequence region of interest. Primers were designed using 

NetPrimer” This would read better as one sentence as the second sentence here is a bit repetitive.
Line 157 – 158: Please be consistent in abbreviations (s or sec).
Line 165: Excessive use of the passive voice here “Double sequence coverage was provided”, please revise.
Line 243: Please remove comma after “thickly” (or change “thickly” to “thick”?)
Line 257: Typo: consist → consists
Line 334: Typo (semi colon)
Line 346: “are influenced by” should be changed to “vary by”
Line 362: y → and?
Line 400: Is the color cream in ethanol throughout, or just the interior? Please confirm.
Line 411: Typo (“holotypewas”)
Line 426: Typo – should be a space in “1mm”
Line 449: Panamá is with an accent here but not in other places in the article; please be consistent one way or the other
Line 497: “This sponge is 
known to harbor a host specific community of Synechococcus spongiarum cyanobacteria”
– could you be more specific here please? I am confused with the terminology as it reads as if you are referring to one species of cyanobacteria as a community. Could you also clarify what you mean by host specific – i.e., is that species of cyanobacteria specific to that species of sponge?
Line 537: “Add your results here”
Line 588: “Throughout” I think implies something more geographically extensive than the sampling conducted here, please change this.
Line 619: Is there a double space here? Hard to tell on the PDF, sorry if not.
Line 627-639: Grammatical mistakes in sentence, please revise
Line 639-641: This last sentence is a little confusingly worded, please revise. I also believe the phrase “make more genomes available” is a bit too passive.
Line 655: “continue” should be “continues”

Table 1 – should this read “number of sigma measured”?

Table S1 does not have a title.

References: There are some inconsistencies and mistakes in your reference list – please go over this carefully and ensure that the style is uniform throughout and remove typos (possibly from referencing software?). E.g., in some references, you have put a space in between the volume and issue number and in some you have not. I have also noted some other mistakes here but it is not necessarily extensive as I only scanned over it:

Line 729: New line missing between references
Line 790: Typo Noth → North. Please also revise the capatilisation in this sentence to match with the rest of the style of your list.
Line 884: Errant comma
Line 891: Volume number is in italics
Sollas 1885 reference – there appears to be a few too many numbers here?
Wiedenmayer 
1977 reference – something strange happened here

Acknowledgements
The contribution numbers are missing here, please add or remove the sentence.

·

Basic reporting

Many suggestions were made for improvement of language, above all as far as punctuation goes. Genbank accession numbers were not included still, but the space for their inclusion is there.

Experimental design

No comments aside those presented in the PDF.

Validity of the findings

There is actually no speculation. But I did suggest some alternative lines of reasoning to be presented, so that the reader can have a more comprehensive view of scenarios other than "we found three new Neopetrosia"

Additional comments

I congratulate the authors on their achieved contribution. Integrative taxonomy is the way to move forward when it comes to cryptc species, and I think the results presented by Vicente et coll is quite convincing of hidden diversity being brought to light. On the other hand, I made a large series of suggestions to avoid ambiguities and inconsistencies as far as arguments go, and a very large series of suggested adjustments to the text. Notorious among the latter is a rather economic use of commas, which renders the text frequently difficult to read. In my opinion the utmost inconsistency lies in the fact that molecular results being presented subsequently in the MS, readers are obliged to accept the new species diagnosed on the basis of absolutely minute morphologic distinctions, even though the first species whose description is presented (N. proxima) is said to be quite variable in many aspects. This is only acceptable if cross-referenced to molecular results, for which reason I suggest to turn the MS "upside-down".

---

## Round 0.2 · accepted · Accept

I was unable to get the reviewers to provide a second look, but on the basis of my own read, I believe the authors have done a good job of addressing the reviewers' comments.

#